# The Chemical Space of Marine Antibacterials: Diphenyl Ethers, Benzophenones, Xanthones, and Anthraquinones

**DOI:** 10.3390/molecules28104073

**Published:** 2023-05-13

**Authors:** José X. Soares, Inês Afonso, Adaleta Omerbasic, Daniela R. P. Loureiro, Madalena M. M. Pinto, Carlos M. M. Afonso

**Affiliations:** 1Laboratory of Organic and Pharmaceutical Chemistry, Department of Chemical Sciences, Faculty of Pharmacy, University of Porto, Rua de Jorge Viterbo Ferreira, 228, 4050-313 Porto, Portugal; jfxsoares@ff.up.pt (J.X.S.); dloureiro@ff.up.pt (D.R.P.L.); madalena@ff.up.pt (M.M.M.P.); 2Interdisciplinary Center of Marine and Environmental Investigation (CIIMAR/CIMAR), Edifício do Terminal de Cruzeiros do Porto de Leixões, Av. General Norton de Matos s/n, 4050-208 Matosinhos, Portugal; 3LAQV-REQUIMTE, Department of Chemical Sciences, Faculty of Pharmacy, University of Porto, Rua de Jorge Viterbo Ferreira, 228, 4050-313 Porto, Portugal

**Keywords:** marine products, natural products, antimicrobial, clustering, drug-like, data visualization, dimensionality reduction

## Abstract

The emergence of multiresistant bacteria and the shortage of antibacterials in the drug pipeline creates the need to search for novel agents. Evolution drives the optimization of the structure of marine natural products to act as antibacterial agents. Polyketides are a vast and structurally diverse family of compounds that have been isolated from different marine microorganisms. Within the different polyketides, benzophenones, diphenyl ethers, anthraquinones, and xanthones have shown promising antibacterial activity. In this work, a dataset of 246 marine polyketides has been identified. In order to characterize the chemical space occupied by these marine polyketides, molecular descriptors and fingerprints were calculated. Molecular descriptors were analyzed according to the scaffold, and principal component analysis was performed to identify the relationships among the different descriptors. Generally, the identified marine polyketides are unsaturated, water-insoluble compounds. Among the different polyketides, diphenyl ethers tend to be more lipophilic and non-polar than the remaining classes. Molecular fingerprints were used to group the polyketides according to their molecular similarity into clusters. A total of 76 clusters were obtained, with a loose threshold for the Butina clustering algorithm, highlighting the large structural diversity of the marine polyketides. The large structural diversity was also evidenced by the visualization trees map assembled using the tree map (TMAP) unsupervised machine-learning method. The available antibacterial activity data were examined in terms of bacterial strains, and the activity data were used to rank the compounds according to their antibacterial potential. This potential ranking was used to identify the most promising compounds (four compounds) which can inspire the development of new structural analogs with better potency and absorption, distribution, metabolism, excretion, and toxicity (ADMET) properties.

## 1. Introduction

Antimicrobial resistance is a major public concern in the 21st century [1]. Due to the overuse and misuse of antibiotics in healthcare and agriculture, conventional drugs used to treat infections are becoming less effective and it is estimated that at least 700,000 people worldwide die each year because of drug-resistant infections [1,2].

Nature has always been a source of inspiration for new drugs and historically, natural products have had a critical impact on the discovery and development of antimicrobials [3]. Nowadays, the impressive potential of natural products [4,5], namely marine natural products (MNPs) [6,7], is being increasingly recognized, as they can provide chemical novelty and novel mechanisms of action. Compounds from different classes are found within MNPs, namely alkaloids, anthraquinones, peptides, polysaccharides, polyketides, and terpenes [8,9]. Among these, marine polyketides (MPs) are a class of secondary metabolites which, due to the versatility of their biosynthetic production mechanism, exhibit a remarkable diversity, both in terms of structural complexity and biological activity [10,11]. Within the different polyketides, anthraquinones (AQs), xanthones (XTs), benzophenones (BZs), and diphenyl-ethers (DEs) are structurally related scaffolds that are present in antibacterial MNPs (Figure 1).

Anthraquinones (AQs) are characterized by a 9,10-anthracenedione core structure, which contains three fused phenyl rings with two carbonyl groups located on the central aromatic ring. Up to 2021, around 700 AQs have been isolated from terrestrial and marine sources, such as fungi, bacteria, lichens, sponges, and marine invertebrates [10]. Xanthones (XTs) are another class of oxygenated heterocycles containing a dibenzo-γ-pyrone moiety [12]. Retaining the tricyclic structure of AQ, XT differs by having one ether and carbonyl linkage instead of two carbonyl groups on the central aromatic ring. Marine XT derivatives have been reported with several biological activities and with suitable features for drug discovery [13,14]. Benzophenones (BZs) are another group of aromatic ketones, which consist of two phenyl rings attached to the same carbon atom of the carbonyl group [15]. Usually, BZs are intermediates in the synthesis of XTs [16]; several BZ derivatives isolated from marine sources exhibit interesting bioactivities [15]. Diphenyl-ethers (DEs) are aromatic compounds that possess two phenyl groups attached by an ether linkage. Similar to the previously reported secondary metabolites, several DEs isolated from the marine environment have shown interesting biological activities [17,18].

In order to successfully become a drug, a compound should occupy a region of the chemical space that guarantees good pharmacodynamic and pharmacokinetic behavior, i.e., the drug-like space. To define the chemical space occupied by a set of compounds, chemical structures are represented through molecular features aimed to encode a certain molecular feature [19]. The molecular features that most affect the pharmacodynamic and pharmacokinetic behavior of a drug are size, flexibility, unsaturation, lipophilicity, polarity, and solubility [20]. Molecular weight (MW) encodes the size of the molecules. The number of rotatable bonds (RB) describes flexibility. The fraction of carbon with sp^3^ hybridization (FCsp3) characterizes saturation. The topological polar surface area (TPSA) represents polarity. The partition coefficient between octanol and water (log P) evaluates lipophilicity, and the logarithm of the solubility measured in mol L^−1^ (log S) evaluates aqueous solubility. Using several descriptors to describe one molecule yields a high dimensional dataset which hampers the attainment of a global perspective over the chemical space. Dimensionality-reduction techniques can increase the interpretability of the dataset by extracting the essential information from the high-dimensional chemical space and projecting it into a lower-dimensional sub-space [21]. Principal component analysis (PCA) is a linear dimensionality-reduction technique suitable for molecular descriptors which preserves the essential parts that contain more variation and removes the non-essential parts with less variation [22].

Molecular fingerprints (FP) are another way to provide a numerical representation of a molecule. Extended connectivity fingerprints (ECFP4) are the most popular fingerprint type and encode the full atom environments up to a diameter of four bonds [23]. By representing a molecule as a bit string, FPs are quite useful to compare the similarity between molecules [24]. FP-based similarity is used by clustering techniques, such as Butina clustering [25], to group similar compounds into clusters. Moreover, FPs can also be used by dimensionality-reduction techniques, such as the TMAP technique. TMAP represents a set of compounds as a map of minimum spanning trees, which allows simultaneous visualization of the global structure of the chemical space and of the local structure of clusters through branches and sub-branches [26].

In this work, we gathered and studied a dataset of 246 marine AQs, XTs, BZs, and DEs. The dataset was analyzed according to the scaffold, antibacterial activity, and marine sources. As far as we know, the chemical space occupied by the studied MPs was mapped for the first time using molecular descriptors and fingerprints and framed according to the drug-likeness concepts and to an antibacterial potential ranking. Figure 2 depicts the schematic representation of the data collection and the tools used for the different calculations and analyses.

## 2. Antibacterial Marine Polyketides

A total of 246 MPs were identified using the bibliographic research described in the Material and Methods section. Of these 246 MPs, 218 (89%) were isolated directly from marine sources, while 28 (11%) were obtained through semi-synthesis from a marine product and by recurring to simple reactions (acetylation [27,28,29], methylation [27,30], dehydration [30,31], or redox reactions [30,31]).

The outer circle of Figure 3a shows the scaffold distribution of the different marine polyketide families and the inner circle shows the distribution between the natural and semi-synthetic derivatives. The anthraquinone scaffold was the most prevalent (119 compounds, 47%), followed by diphenyl ether (96 compounds, 38%), xanthone (25 compounds, 10%), and benzophenone (13 compounds, 5%). A total of 7 dimeric molecules, containing more than one type of scaffold, were identified and they were classified in both scaffolds. Semi-synthetic derivatives were only identified in the anthraquinone and diphenyl ether groups. This can be related to the higher prevalence of these scaffolds when compared to xanthones and benzophenones.

The inner circle of Figure 3b shows the distribution of the MPs isolated from marine associations, such as fungi and bacteria. In this case, the active compound was considered to be produced by the bacteria. Five MPs were isolated from more than one source (Appendix A).

Figure 3c shows the distribution of the genus of marine organisms. Considering MPs isolated from fungi, *Aspergillus* (75 MPs in total) was the most frequent genus, followed by *Penicillium* (16 MPs in total). Considering MPs isolated from bacteria, *Streptomyces* was the most frequent genus (27 MPs). Considering MPs isolated from sponges, *Lamellodysidea herbacea* was the most frequent species (13 MPs).

Figure 3d shows the geographic distribution of the MPs around the globe. The locations displayed in the figure refer to the sea (for example, the South China Sea) or to the country (for example, Canada) where the MPs were isolated. Most MPs were isolated in the region between the Tropic of Cancer and the Tropic of Capricorn, with special incidence in Southeast Asia. China was the country from which the most MPs were isolated (102 MPs in total), namely from the Qingdao region (25 MPs), Dalian region (19 MPs), and Hainan province (14 MPs).

## 3. Molecular Features of Antibacterial Marine Polyketides

The most important molecular features in Medicinal Chemistry are size, flexibility, unsaturation, lipophilicity, polarity, and solubility. For each MP, molecular descriptors that encode these molecular features were calculated (Appendix A) and analyzed according to their scaffold families (Figure 4).

The molecular size, which is usually evaluated by the MW of the molecule, mostly influences the interaction with the pharmacological target and the permeability across the membrane [32]. The molecular mass should be under 500 g mol^−1^ for optimal interaction with a pharmacological target and good membrane permeability [33]. Most of the identified MPs ranged between 300 and 600 g mol^−1^ (Figure 4a). DE and XT were associated with a wide range of molecular sizes (interquartile range (IQR) of 316.3 for XT and 191.3 for DE). The wide range of DE and XT sizes is due to the presence of dimeric compounds and, in the case of DE, also to four outliers containing seven bromine atoms in their composition (see Appendix A for examples). AQ is similar in terms of molecular size to XT (mean value of 399.4 and 413.9 for AQ and XT, respectively), but the dimeric AQs are clearly identified as outliers (grey diamonds in Figure 4a). BZs have the smallest molecular size (mean value of 356.7) and have the narrowest size distribution (IQR of 56.7). However, it should be mentioned that BZ is the class with the lowest number of compounds and, consequently, with the lowest probability of having different-sized compounds.

The molecular flexibility can be inferred by the number of rotatable bonds (RB). High flexibility is a characteristic of natural products [34] and is associated with more favorable drug-like properties and with lower ability to act with multiple molecular targets [35]. This feature has been clearly reflected in approved drugs. The drugs approved between 1998 and 2017 have a statistically higher number of rotatable bonds than those approved between 1990 and 1997 [36]. Intrinsically, the core scaffold of each of the four studied families contains the same number of rotatable bonds (two RB). Therefore, the flexibility found in MPs is a consequence of their substitution patterns (number and type). Most of the reported MPs have fewer than five rotatable bonds (Figure 4b), highlighting their rigidity. AQ and XT are the most rigid compounds (median of one and two, respectively), while DE and BZ are the most flexible (median of four each).

Unsaturation of the molecules can be evaluated by the fraction of sp3 carbons (FCsp3). FCsp3 also provides information regarding the complexity of the structure [37,38] and affects several ADMET parameters, such as aqueous solubility, permeability, plasma protein binding, and hERG and CYP3A4 inhibition [35]. Most of the reported MPs have a high degree of unsaturation (FCsp3 < 0.3) (Figure 4c). DE is the class of the studied compounds with the lowest FCsp3 carbons (mean of 0.14). FCsp3 and flexibility are often directly correlated (a high FCsp3 usually corresponds to high flexibility). However, DEs show a low FCsp3 and, simultaneously, a high RB. This can be explained by the occurrence of dimeric structures, which are essentially unsaturated and formed by monomers linked by rotatable ether linkages. AQ and XT classes have similar FCsp3 values (means of 0.27 and 0.26 for AQs and XTs, respectively), while the BZ class shows a narrower distribution (IQR of 0.04).

Lipophilicity can be measured by the partition coefficient of the substance between *n*-octanol and water (log P) and deeply affects both the pharmacokinetics and pharmacodynamics [33,39]. Several methods can be used to estimate log P [40]. However, different methods can lead to different log P values for the same molecule [41]. In this work, six in silico methods (MolLogP, iLOGP, XLOGP3, WLOGP, MLOGP, Silicos-IT Log P) were used to predict the log P value of each compound (Appendix A) and the obtained mean log P value was considered for the analysis of the classes of compounds depicted in Figure 4d. DE stands out from the other compound classes as the most lipophilic class of MPs (mean of 4.68), attesting to the presence of several lipophilic halogen atoms (bromine and chlorine) and the absence of polar groups, such as hydroxyl groups (Figure 4d). The remaining classes have similar log P values (median of 2.25, 2.65, and 2.51 for AQs, XTs, and BZs, respectively) with different ranges of value distribution.

Molecular polarity can be evaluated by the sum of topological surface areas of polar atoms in a molecule (TPSA) and affects the bioavailability of drugs [42,43]. Molecules with a polar surface area of greater than 140 Å^2^ tend to be poor at permeating cell membranes. DE is the class with the lowest molecular polarity (median of 54.3), while the remaining classes have similar mean values (median of 115.1, 96.2, and 113.3, for AQs, XTs, and BZs, respectively) (Figure 4e). When compared with the log P value, DE is simultaneously the class with the lowest molecular polarity and the highest lipophilicity. XT shows a wide distribution of TPSA values and a narrow distribution of log P values. High and low polar XT compounds present similar lipophilicities, showing that the presence of polar groups in XT, such as the carbonyl group, does not affect drastically the lipophilicity of XTs (Figure 4d,e).

Water solubility can be expressed as the 10-based logarithm of solubility (log S). Water solubility is very important to ensure that a drug reaches its therapeutic concentration. Low aqueous solubility is the major problem encountered with the formulation development of new chemical entities [44]. Any orally administered drug must be present in the form of a solution at the site of absorption. Compounds considered to have a poor water solubility, with a log S lower than −4, tend to not be orally bioavailable [44]. The log S values for all of the MPs compounds were predicted in silico by two different methods (Appendix A), and the mean log S values were considered for the analysis (Figure 4f). The majority, 82% of the studied MPs, can be classified as practically insoluble with a log S < −4. The most soluble MPs were AQs, followed by XTs = BZs and DEs (the less soluble). This behavior can be justified by the size (Figure 4a), lipophilicity (Figure 4d), and polarity (Figure 4e) of the molecules of each class of compounds. DE is the class with a higher molecular weight, higher lipophilicity, and lower polarity than the remaining classes and this is reflected by having the lowest water solubility. AQs show the opposite characteristics. It should be highlighted that the analyzed log S values did not consider the pH-dependent solubility. In fact, several MPs possess ionizable groups, such as carboxylic acid (34 MPs) and amine groups (8 MPs), which have pH-dependent log S values. This is relevant because drugs within the organism can show different solubility characteristics, depending on the pH of the body compartment they are traveling within.

PCA analysis was performed to reduce the complexity of the chemical space, composed of six features, and transform it into an easier-to-interpret chemical space.

As each feature is expressed in different units and presents different magnitudes, before the PCA analysis, the dataset was normalized into the same scale. Seven different scaling/transformers were studied which have different impacts on the distribution of the scaled data (Appendix A). “Standardization” is a common technique for scaling a dataset to PCA. As this technique centers the values around the mean with a unit standard deviation, the presence of outliers, as in our case, may distort PCA results (Appendix A). In the present context, outliers are perceived as exceptions that should be retained, but should not have a decisive impact on the analysis. Among the studied techniques, the “robust scaler” was selected for this analysis, due to the efficiency shown in the scaling of our dataset and due to its robustness to the presence of outliers.

As depicted in Figure 5a, the first principal component (PC1) and the second principal component (PC2) account for most of the variation. Using only PC1 and PC2 is enough to provide a good approximation of the chemical space occupied by MPs, because they account for more than 80% of the variation in the data (Figure 5b). The loadings of PC1 and PC2 revealed the contributions of the original features to these new principal components (Figure 5c). The log P and log S data information was captured by PC1. TPSA and FCSp3 data information was captured by PC2. MW and RB data information was captured by both PC1 and PC2. The analysis of the direction of the vectors in the loadings plot allows us to infer the correlation between the different features (Figure 5c, orange arrows). Two features are positively correlated when the two respective vectors are close to each other, and they are negatively correlated when far away. The analysis confirmed some expected correlations: (i) the negative correlations between lipophilicity (log P) and water solubility (log S), between water solubility (log S) and size (MW), and between lipophilicity (log P) and polarity (TPSA); and (ii) the positive correlations between water solubility (log S) and polarity (TPSA), and between size (MW) and rotatable bonds (RB). The analysis also showed two unexpected correlations: the negative correlation between the number of rotatable bonds (RB) and water solubility (log S), and the positive correlation between polarity (TPSA) and the extension of unsaturation (FCsp3). Usually, higher free movements of the molecule (high RB and flexibility) lead to higher solubility. This unforeseen negative correlation shows that, for the studied MPs, size (MW) has a greater influence on solubility than flexibility (RB). Thus, a high RB in this case means large molecules (high MW), rather than highly flexible molecules (high RB). Usually, polarity (TPSA) is not directly related to molecular unsaturation (FCsp3). However, for the studied MPs, high values for polarity (TPSA) are associated with a low molecular unsaturation (FCsp3). Polar groups in MPs commonly result from the addition of polar groups to unsaturated bonds, such as hydration reactions, which can explain the verified results. Using the selected chemical space occupied by the MPs, each MP was projected and labeled according to the chemical class scaffolds (Figure 5d). AQ is the most heterogeneous class of all the MPs, showing the widest variation in the two-dimensional space. The DE class tends to form two well-separated clusters: one that places some DE (orange) structures with similar features to AQs (blue) and XTs (green) and another one that is more specific and different from the other scaffold features. XTs (green) and BZs (red) present more homogeneous and close feature values. Nevertheless, it should be kept in mind that these two classes are less represented than AQ or DE.

## 4. Clustering Analysis of Antibacterial Marine Polyketides

The cluster analysis performed with PCA is based on molecular descriptors that use molecular and physico-chemical properties to encode chemical information. However, the studied molecular descriptors do not directly encode the chemical structure. In order to address this potential pitfall, molecular fingerprints were used as surrogates of chemical structures in a Butina clustering analysis.

For each MP, the ECFP4 values were calculated using Morgan implementation of RDkit. To avoid bit collision, when two features were mapped in the same bit position, the number of unique fingerprints was evaluated using different bit lengths (Appendix A). The bit length of 16,392 was selected, as it corresponds to the bit length where the number of unique fingerprints plateaued.

Butina clustering is based on the Tanimoto coefficients, which measure the similarity between the fingerprints of two compounds [25]. The clustering is performed by grouping similar molecules, i.e., molecules with similar Tanimoto coefficients. The dataset is sorted in descending order of the number of similar compounds. The first one, the compound with the largest number of similar compounds, is considered the cluster centroid, and its pairwise Tanimoto coefficient to all other compounds is calculated. All those molecules with a Tanimoto coefficient higher or equal to a defined threshold value are attributed to a distinct cluster. This process repeats until all compounds have been attributed to a distinct cluster. The threshold value is the most important parameter for the Butina clustering. The higher the threshold, the more compounds are considered similar, and the set is clustered into fewer clusters. The lower the threshold, the fewer compounds are considered similar, and the set is clustered into more clusters. Different threshold values were evaluated (Appendix A). Among the evaluated thresholds, the threshold value of 0.5 was selected because: (i) it guarantees that within each cluster, at least half of the fingerprints are equal; (ii) the cluster sizes have a smooth distribution; (iii) the number of singletons, clusters with only one element, is not extreme. Applying the threshold value of 0.5, a total of 76 clusters were obtained (Figure 6a). Each cluster corresponds to a class of similar substances. The first cluster is the most representative and comprises 22 compounds (corresponding to 9% of the set). Several clusters contained 3 or fewer compounds (cluster 19 to cluster 76) and these clusters were presented as a miscellaneous class (corresponding to 35% of the set) (Figure 6a, black bars).

For each cluster, the centroid corresponds to the compound that was considered the archetype of that cluster. The structures of the centroids of clusters 1, 2, 3, and 4 are depicted in Figure 6b–e (on the left). The structures of the most dissimilar compounds within these are also depicted in Figure 6b–e (on the right). Similarity maps between the centroid and the most dissimilar compound are also depicted in Figure 6b–e (on the right) [45]. In these similarity maps, the atoms colored in green have a positive impact on the similarity, while the atoms colored in pink have a negative impact on the similarity. These clusters were used to analyze the antibacterial activity of marine polyketides.

## 5. Antibacterial Activity of Marine Polyketides

The antibacterial activity of the studied marine polyketides was searched, identified, and collected systematically from the literature (Appendix A).

The most assayed substances for antibacterial activity were AQs, followed by DEs, XTs, and BZs (Figure 7a). Frequently, each compound was screened against several bacteria, involving several antibacterial assays. Consequently, the number of identified assays was bigger than the number of the studied marine polyketides. From the different polyketide families, XTs revealed the greatest number of compounds with antibacterial activity (62%), followed by BZs (58%), AQs (47%), and DEs (41%) (Figure 7a, dark blue bar). Interestingly, roughly half of the polyketides exhibited antibacterial activity. These data prove the importance and great potential of polyketides as antibacterial hits.

Most of the assays described in the literature have been performed against Gram-positive bacteria (Figure 7b). Considering the performed assays against Gram-positive bacteria, 41% showed bacterial growth inhibition. Considering the performed assays against Gram-negative bacteria, 50% showed bacterial growth inhibition (Figure 7b). Figure 7c shows the most common bacteria screened with the identified marine polyketides. These bacteria are part of the panel of bacteria that are commonly used for screening antibacterial activity.

Despite the promising antibacterial activity reported for MPs, their mode of action has not been investigated in detail. Bearing in mind the structural similarity between these marine polyketides and terrestrial or synthetic polyketides, it is reasonable to assume that they could share similar modes of action. Natural and synthetic polyketides bearing the four studied scaffolds have been shown to act on efflux pumps [46], bacterial cell membranes [46,47], DNA gyrase B [48], and transpeptidases [49] and through reactive oxygen species [50]. As the studied MPs can show different mechanisms of action, we focused our study on the de facto expressed antibacterial activity. However, the antibacterial activity of the studied MPs is reported in the literature using different metrics and units (Figure 7d). Minimal inhibitory concentration (MIC) is the most used metric and corresponds to the lowest concentration of an antibacterial that will inhibit the visible growth of a bacterium. The diameter of the zone of inhibition (DIZ) determined by the disk diffusion method and the concentration that inhibits the growth of half of an inoculum (IC_50_) are the other two used metrics. These metrics are often used indiscriminately with the analyzed polyketides, which hampers the comparison and discussion of the obtained data. Moreover, even when the used metric is the same, the obtained results are expressed in different units. For instance, considering the analysis of MIC values expressed as mg mL^−1^ can be deceiving when compared with the same MIC values expressed as µM. This is especially relevant with polyketides, which have a wide range of molecular weights (Figure 4c). In order to provide the most valuable discussion, all the MIC values were converted to the same unit, the log of their micromolar concentration (log MIC in µM). A similar approach was followed for the DIZ data (Appendix A). Figure 7e points out the distribution of the reported log MIC values according to the MP scaffold families, showing similar mean values for all families. This information can be interpreted as demonstrating that all of the families have a similar potency as antibacterial agents and that the scaffold by itself cannot be used as a predictor for antibacterial potency. However, it is possible to define groups of compounds with different levels of potency, using the previously discussed clusters. FP-based clusters can discriminate clusters with different levels of potency (Figure 7f). For example, clusters 1 and 9 contain active compounds, while clusters 11 and 12 do not contain any active compounds. As expected, MPs attributed to the miscellaneous class shows a wide distribution of log MIC values. As similar compounds, i.e., belonging to the same cluster, tend to have a similar antibacterial activity, it is possible to infer the existence of a relationship between the structure and the antibacterial activity.

The antibacterial activity of an MP varies between assays. Depending on the type of bacteria, the type of strain, or the type of assay, the same compound can be associated with completely different MIC values. These variations are normal, but they hamper the evaluation of the potential of a certain MP to become a lead compound. In order to assess the antibacterial potential of the studied MPs, two activity thresholds, a log DD of 3 and a log MIC of 1, were considered (see the Appendix A for further explanation of the segmentation). Based on these thresholds, the studied MPs were segmented into four different categories: (i) not active, corresponding to MPs that were inactive in all performed assays; (ii) not promising, corresponding to MPs which, despite being active in some assays, never reached the activity threshold; (iii) promising, corresponding to MPs which depending on the assays exhibited activity above or below the thresholds; (iv) very promising, corresponding to MPs that always exhibited activity above the threshold. Of the 246 MPs studied, 94 were classified as not active, 57 were classified as not promising, 65 were classified as promising, and 30 were classified as very promising.

In order to shed some light on the relationship between the structure and the antibacterial activity, the TMAP algorithm was applied to generate a two-dimensional visualization of the chemical space occupied by the identified MPs (Figure 8). The obtained tree-based layout allows the simultaneous visualization of the global structure of the chemical space, through the stem, and of the local structure of clusters, through the branches and sub-branches. Within this representation, each data point corresponds to an MP and the closer the data points are, the more molecular fingerprints they share. In order to visualize the molecular structures associated with each point, an interactive version is available upon request. AQs were in the upper section of the tree-map (Appendix A). DEs were in the lower section of the tree-map. BZs and most XTs were located between the two sections of the tree-map.

The TMAP visualization was colored according to the categories of antibacterial potential. MPs classified as very promising were not concentrated in a specific region of the chemical space but formed small islets in different positions. The spread over the different regions of the chemical is attributed to the broad notion of biological activity used for the classification. In this analysis, we are classifying compounds without detailing the differences between the assayed bacteria. Therefore, very different molecular traits, corresponding to well separated positions in the TMAP projection, can exhibit different mechanisms of action. Most of the very promising MPs are in the upper section, corresponding to AQs, and in the lower section, corresponding to DEs. Bearing this in mind, it is reasonable to assume that at least two different mechanisms of action are present for these compounds. The fact that very promising MPs tend to form small islets in the chemical space means that they share common molecular traits, which proves the existence of a link between the structure and their antibacterial activity. However, not all similar compounds, i.e., compounds that form a branch on TMAP projection, were classified as very promising. Subtle nuances in the chemical structure led to differences in the antibacterial activity described for the MPs.

From a medicinal chemistry perspective, the very promising MPs are the most interesting. These MPs always exhibited a log MIC lower than one (which corresponds to a MIC value of 10 µM) and in some cases even lower than zero. In addition to potency, the drug-likeness of these compounds was also quantitatively analyzed. The quantitative estimate of drug-likeness (QED) expresses drug-likeness on a scale ranging from zero (when all properties are unfavorable) to one (when all properties are favorable) [51]. Figure 9 depicts the structures and QED values of the MPs classified as very promising, which are sorted according to the structural sub-family.

The tricyclic polyketides subfamily contains AQs or AQs derivatives bearing a simple substitution pattern. These compounds are the smallest compounds within this series (MW < 300). In common, these structures possess a substituent (hydroxyl or methyl group) at both ortho positions to the carbonyl group. The tetracyclic subfamily contains AQs or AQs derivatives with at least one additional heterocycle. These compounds are the least lipophilic compounds within this series (mean log P of 1.25) and are the most drug-like molecules (mean QED value of 0.63). The prenylated subfamily contains MPs with two aryl groups, linked either by an ether or a carbonyl linkage, substituted with a prenyl group. These compounds are non-planar and have the highest number of RBs within this series (six rotatable bonds). The bisanthraquinones subfamily contains two dimeric AQs, produced by a marine streptomycete, and three semi-synthetic derivatives obtained through dehydration, oxidation, and reduction. These large compounds (mean MW of 554.53) are the most polar compounds within this series (mean TPSA of 154.63). Bisanthraquinones are the least drug-like molecules (mean QED value of 0.26). The polybrominated diphenyl ethers subfamily contains DEs substituted with several bromine atoms (from 3 up to 7 bromine atoms). These compounds are the most lipophilic (mean log P of 5.70) and least polar (mean TPSA of 47) within this series. All of them are practically insoluble in water (mean log S of −7.34).

## 6. Materials and Methods

The literature survey was executed using Scopus (https://www.scopus.com/, accessed on 10 March 2023), Web of Science (https://www.webofscience.com, accessed on 10 March 2023), and Google Scholar (https://scholar.google.com accessed on 10 March 2023) considering original research papers published up to June of 2022. The keywords used were: “marine AND anthraquinone*”, “marine AND xanthone*”, “marine AND benzophenone*”, and “marine AND diphenyl ether AND (category: environment sciences)”. Antibacterial activity data were collected from the reference describing isolation or from the original reference (when the MPs had been previously isolated and assayed).

Molecular descriptors were calculated using the RDkit (release 2021_03_5 Q1 2021, https://www.rdkit.org/ accessed on 1 March 2023) and SwissADME [20]. Data analysis was performed in Python 3.8.16, using the RDkit 2021.03.5, Pandas 1.5.2, Numpy 1.23.5, Matplotlib 3.7.0, Seaborn 0.12.2, and Sci-kit learn 1.2.1 libraries. These tools were installed through Conda (https://docs.conda.io/projects/conda, accessed on 3 January 2023). The code for the analysis is accessible at https://github.com/jxsoares/antibacterial-marine-polyketides. The chemical structures were drawn with MarvinSketch 22.22 (https://chemaxon.com accessed on 20 November 2022).

## 7. Conclusions

The marine environment is an important source of natural polyketides with antibacterial activity. A total of 246 MPs were isolated from marine sources bearing an anthraquinone, xanthone, benzophenone, or diphenyl ether scaffold, with anthraquinone and diphenyl ether being the most prevalent scaffolds. MPs were mainly isolated from marine fungi collected between the Tropic of Cancer and the Tropic of Capricorn.

In general, the studied MPs can be defined as unsaturated, inflexible, and water-insoluble molecules. DE are very lipophilic and non-polar molecules, while the remaining MPs exhibit moderate lipophilicity and polarity. The majority of the identified MPs were above the fragment-like size threshold (MW < 300) but below the drug-like size threshold (MW < 500). PCA of the studied molecular descriptors revealed negative correlations between log P and log S, between log S and size, and between log P and TPSA. PCA also showed positive correlations between log S and TPSA and between MW and RB.

The identified MPs have been sorted into clusters according to their molecular similarity. The large number of clusters obtained using a loose threshold (threshold of 0.5 in the Butina clustering algorithm) highlights the large structural diversity present within this dataset. The analysis of natural products is usually performed with scaffold-based classification. However, our cluster-based classification, which is based on molecular fingerprints rather than on scaffold description, provided better discrimination between active and non-active compounds than the conventional scaffold-based classification. The large chemical diversity present in the dataset was also evidenced by the TMAP, which shows the common root between all MPs and the numerous spanning branches encoding the different chemical structures. Coloring the TMAP map with our antibacterial potential ranking enabled the easy visualization of the regions of the chemical space containing the most promising compounds. Within the very promising compounds, four tricyclic polyketides were identified which exhibit good antibacterial activity and good drug-like properties. It should be mentioned that in this analysis, we considered a broad notion of antibacterial activity based only on MIC values. The compounds identified as very promising might have different mechanisms of action and/or different potency for different bacteria strains. Independently of the intricacies of the antibacterial activity, these compounds constitute promising starting points for the design and synthesis of novel molecules which could expand our understanding of the antibacterial action of these marine natural products.

## Figures and Tables

**Figure 1 molecules-28-04073-f001:**
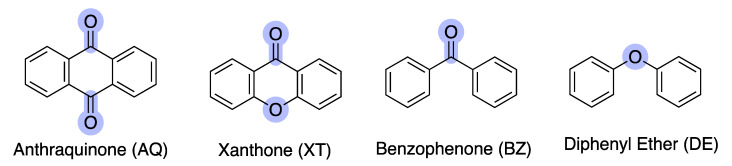
Polyketide scaffolds present in the marine environment: anthraquinone (AQ), xanthone (XT), benzophenone (BZ), and diphenyl ether (DE). The differences between scaffolds are highlighted in light purple.

**Figure 2 molecules-28-04073-f002:**
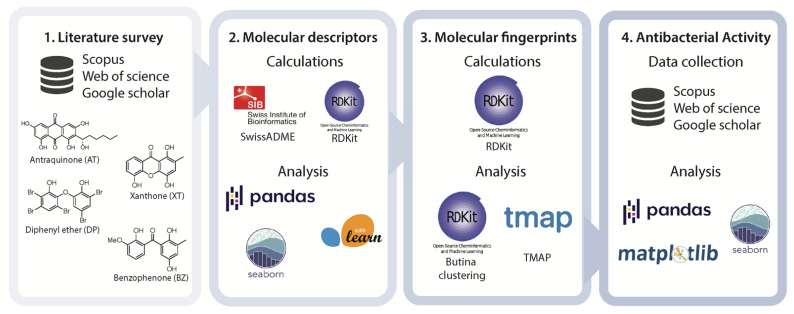
Schematic representation of the data collection and processing.

**Figure 3 molecules-28-04073-f003:**
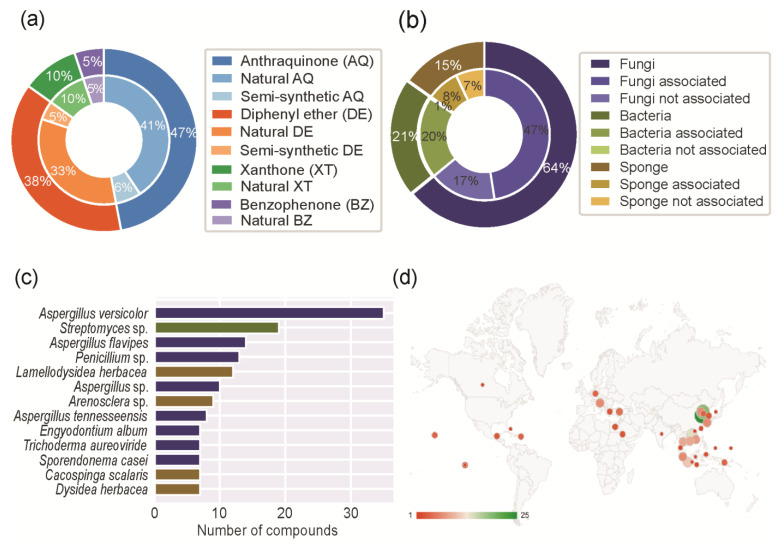
Chemical (**a**), genus (**b**), species (**c**), and geographic distribution (**d**) of the reported MPs.

**Figure 4 molecules-28-04073-f004:**
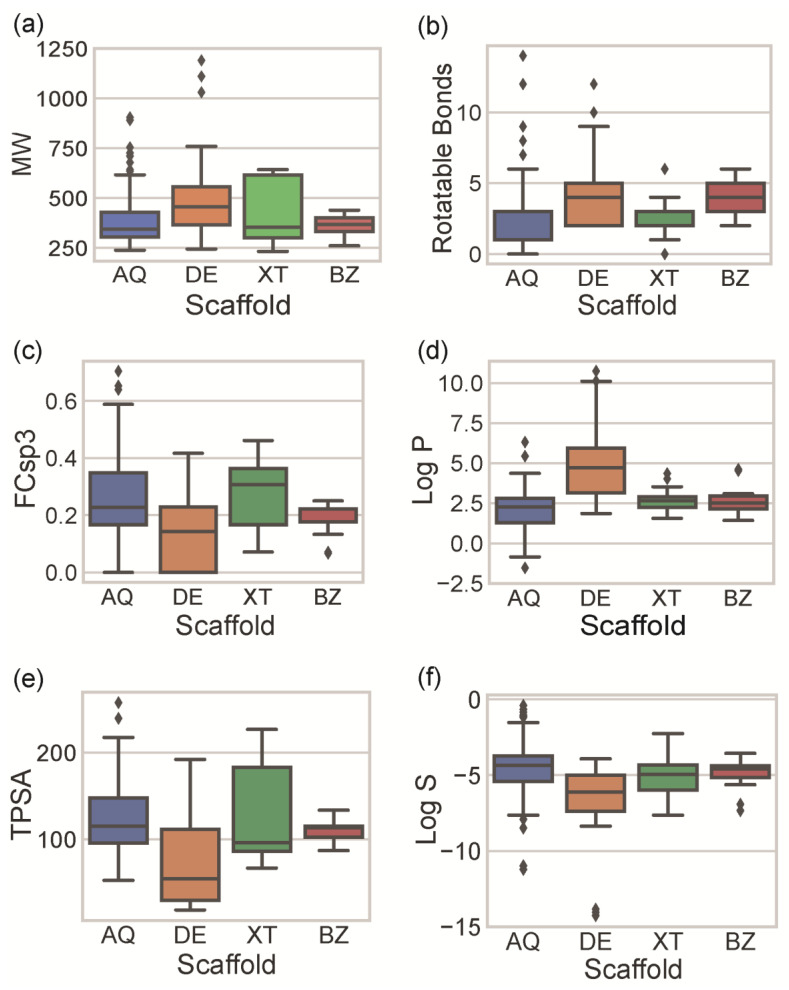
Distribution of MW (**a**); number of RBs (**b**); FCsp3 (**c**); log P (**d**); TPSA (**e**); and log S (**f**) according to the scaffold of the identified marine polyketides (blue for AQ, orange for DE, green for XT, and red for BZ). The box represents the values between the interquartile range (first and third quartiles). The whiskers represent the points within 1.5 times the interquartile range, and the diamond markers the outliers.

**Figure 5 molecules-28-04073-f005:**
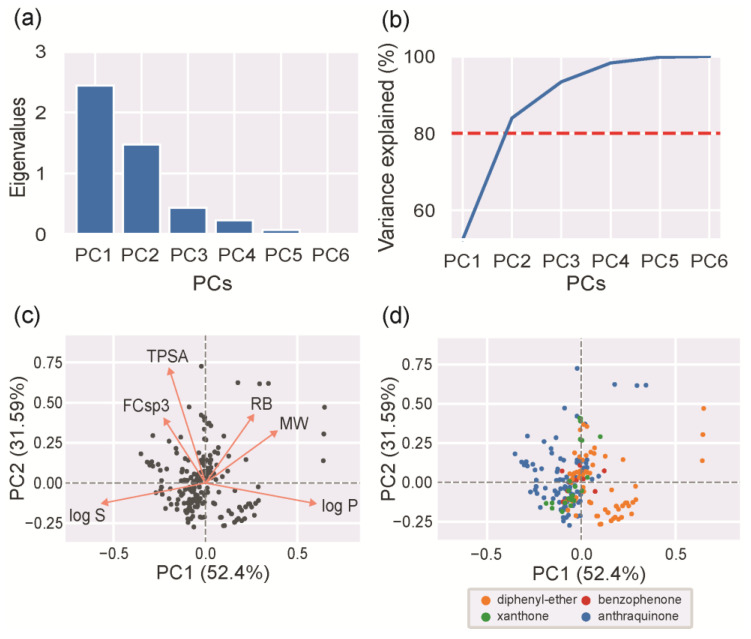
PCA analysis: (**a**) eigenvalues and (**b**) cumulative percentage of explained variance for each principal component (red dashed line represents the threshold); (**c**) loadings plot with the first (PC1) and second (PC2) components with projections of each MP in the reduced space (black circles); (**d**) PCA scatter plot with projections of each MP in the reduced space, where each MP was colored according to its scaffold (blue for AQs, orange for DEs, green for XTs, and red for BZs).

**Figure 6 molecules-28-04073-f006:**
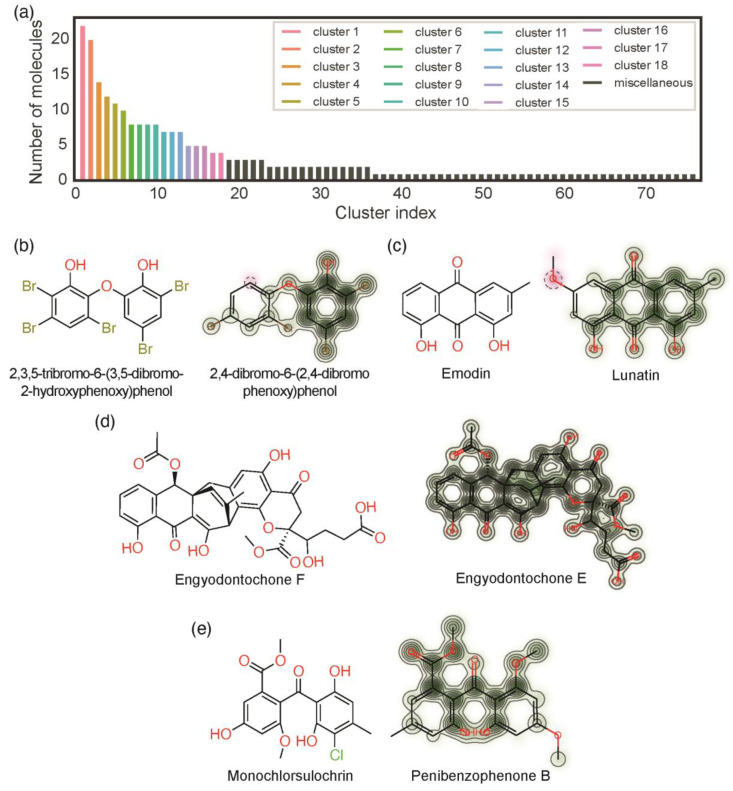
Clusters analysis: (**a**) number of clusters and number of compounds per cluster; chemical structure of the centroids the similarity map between the centroid and the most dissimilar compound within a cluster of DEs (**b**), AQs (**c**), XTs (**d**), and BZ (**e**).

**Figure 7 molecules-28-04073-f007:**
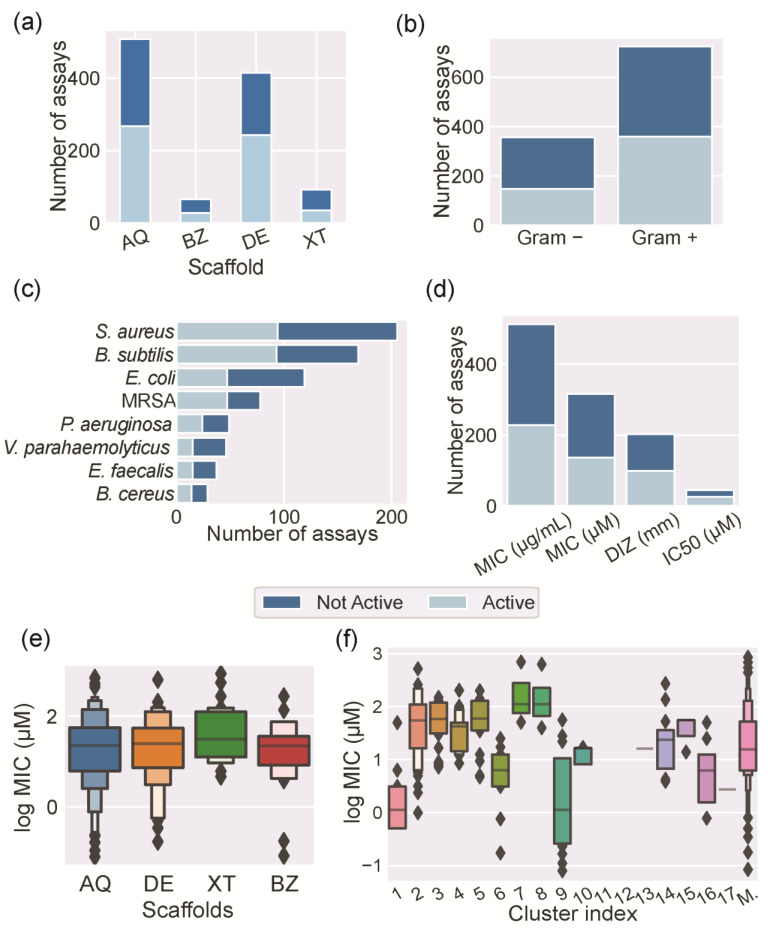
Antibacterial activity analysis: (**a**) bar plot of the antibacterial assays performed for the studied MPs according to their scaffold; (**b**) bar plot of the antibacterial assays according to the Gram classification of the screened bacteria; (**c**) bar plot of the most frequently screened bacteria; (**d**) bar plot of the metrics and units used to evaluate the antibacterial activity; box plot of the distribution of log MIC values according to the (**e**) scaffold or (**f**) cluster of the MPs.

**Figure 8 molecules-28-04073-f008:**
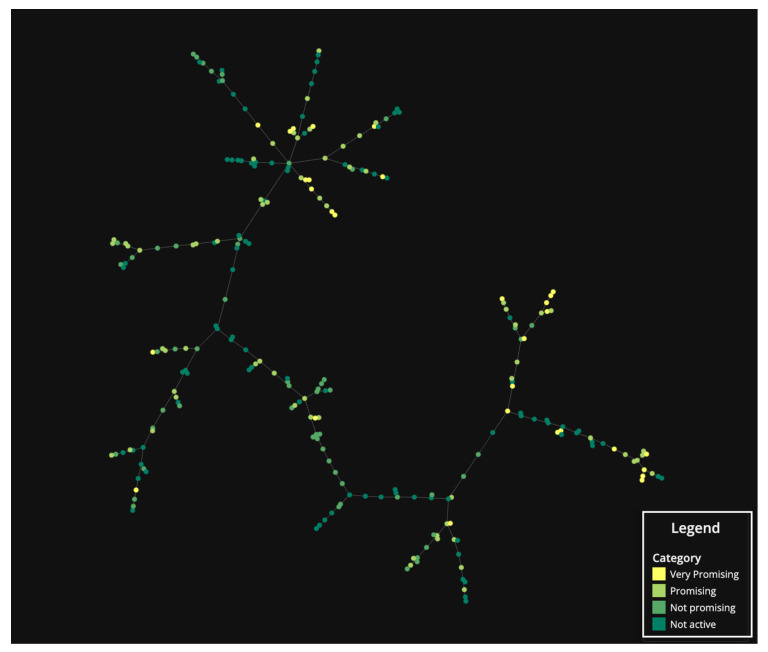
TMAP visualization of the studied MPs colored by the category of antibacterial potential (not active, not promising, promising, very promising).

**Figure 9 molecules-28-04073-f009:**
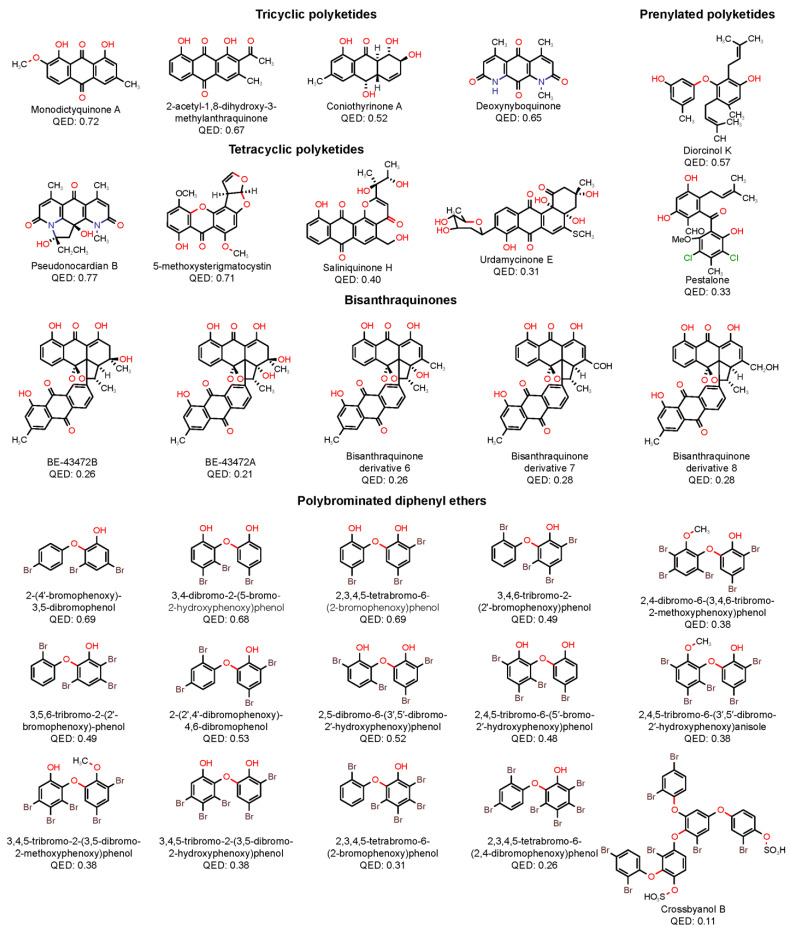
Structures and QED values of the very promising MPs sorted according to the structural sub-family.

## Data Availability

The data presented in this study are available in the Appendix A and at https://github.com/jxsoares/antibacterial-marine-polyketides.

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
