# Peer review of "The Chemical Space of Marine Antibacterials: Diphenyl Ethers, Benzophenones, Xanthones, and Anthraquinones"

_molecules, 2023, doi:10.3390/molecules28104073_

Round 1

Reviewer 1 Report

In this manuscript, the authors reported the exploration of the chemical space of marine polyketides as antibacterials. Molecular descriptors and fingerprints were used to characterize the molecules, and principal component analysis was performed to identify the relationships among the different descriptors. Presently, antibiotic resistance is an increasingly serious problem and marine natural products are tremendous resources of novel antibacterials. Thus, this work is interesting and worthy of publication. Nevertheless, only 246 molecules have been included in the study and the number is too small for chemical space exploration. The authors are encouraged to collect and investigate more complex compounds, such as polyketides Turbinmicin (ref: Science, 2020, 370(6519): 974-978).

In conclusion, I recommend its publication after major revision.

Author Response

At first, we would like to express our gratitude to you for the time dedicated to our manuscript and the valuable comments which improved the quality of our paper. The manuscript was thoughtfully revised and modified according to the reviewer's recommendations.

Reviewer 1 comment´s:

“In this manuscript, the authors reported the exploration of the chemical space of marine polyketides as antibacterials. Molecular descriptors and fingerprints were used to characterize the molecules, and principal component analysis was performed to identify the relationships among the different descriptors. Presently, antibiotic resistance is an increasingly serious problem and marine natural products are tremendous resources of novel antibacterials. Thus, this work is interesting and worthy of publication. Nevertheless, only 246 molecules have been included in the study and the number is too small for chemical space exploration. The authors are encouraged to collect and investigate more complex compounds, such as polyketides Turbinmicin (ref: Science, 2020, 370(6519): 974-978).
In conclusion, I recommend its publication after major revision.”

Author´s response: We thank the reviewer for this observation. Following the reviewer's suggestion, we search for compounds that were not present in the first version of the manuscript. Within the time period defined for the literature survey, were not able to collect more compounds. The compound suggested by the reviewer, Turbinmicin, is an antifungal compound that is out of the scope of the present work (ref: 10.1126/science.abd6919; ref: 10.1172/JCI145123; ref: 10.3389/fmicb.2022.911322). It should be mentioned that the number of identified polyketides is only referent to the defined time period and that new compounds were reported in the last couple of months. In the new version of the manuscript, this aspect has been clarified.

Revised manuscript: (line 459) "The literature survey was executed using Scopus (https://www.scopus.com/ accessed on 10 March 2023), Web of Science (https://www.webofscience.com accessed on 10 March 2023), and Google Scholar (https://scholar.google.com accessed on 10 March 2023) considering original research papers published up to June of 2022"

Reviewer 2 Report

This article presented antibacterial assessment against different bacterial strains by various polyketides isolated from marine source. In addition, the study identified 246 molecules bearing an anthraquinone, xanthone, benzophenone, or diphenyl ether scaffold. The study is well organized and data is well arranged. The findings would be helpful for future studies. Before recommending this article for publication, there are some shortcomings for that should be resolve.

Line 18. evolution, namely, to be antibacterial agents” the sentence is not clear grammatically.

Elaborate abbreviations at first use.

The abstract is only focused on what the authors did like methodology. No specific findings are presented in the abstract.

Line 43 should be cited with recent studies. The following studies would be helpful https://doi.org/10.3390/microorganisms10050954, https://doi.org/10.3390/molecules27196281

In introduction novelty of the study should be explained.

Line 107 correct the sentence.

Materials and methods do not cover complete details. The authors are directed to provide proper details with links of the software and databases.

Also provide scheme or graphical representation of the materials and methods data collection.

The study lack discussion.

Conclusion looks like summary.

Conclusion should be based on results providing study gap and future perspective.

Some sentences should be revise. specifically long and unclear sentences. 

Author Response

At first, we would like to express our gratitude to you for the time dedicated to our manuscript and the valuable comments which improved the quality of our paper. The manuscript was thoughtfully revised and modified according to the reviewer's recommendations.

Reviewer 2 comment´s:

This article presented antibacterial assessment against different bacterial strains by various polyketides isolated from marine source. In addition, the study identified 246 molecules bearing an anthraquinone, xanthone, benzophenone, or diphenyl ether scaffold. The study is well organized and data is well arranged. The findings would be helpful for future studies. Before recommending this article for publication, there are some shortcomings for that should be resolve.

Line 18. evolution, namely, to be antibacterial agents” the sentence is not clear grammatically.

Author´s response: We would like to thank the reviewer for this comment. In the new version of the manuscript, this sentence was clarified.

Revised manuscript: (line 16) "Evolution drives the optimization of the structure of marine natural products to act as antibacterial agents."

Elaborate abbreviations at first use.

Author´s response: We would like to thank the reviewer for this comment. In the new version of the manuscript, all abbreviations were elaborated at first use.

Revised manuscript: line 30: "(…) using the tree map (TMAP) unsupervised machine learning method."

line 34: "(…) absorption, distribution, metabolism, excretion, and toxicity (ADMET) properties."

line 276: "The cluster analysis performed with principal component analysis (PCA) (…)."

The abstract is only focused on what the authors did like methodology. No specific findings are presented in the abstract.

Author´s response: We understand the reviewer´s comment. In the new version of the manuscript, the findings identified in our study were incorporated in the abstract.

Revised manuscript: (line 24) "Generally, the identified marine polyketides are unsaturated and water-insoluble compounds. Among the different polyketides, diphenyl ethers tend to be more lipophilic and non-polar than the remaining classes."

(line 27) "A total of 76 clusters were obtained, with a loose threshold for the Butina clustering algorithm, highlighting the large structural diversity of the marine polyketides. The large structural diversity was also put in evidence in the visualization trees map assembled using the tree map (TMAP) unsupervised machine learning method."

Line 43 should be cited with recent studies. The following studies would be helpful https://doi.org/10.3390/microorganisms10050954, https://doi.org/10.3390/molecules27196281

Author´s response: We would like to thank the reviewer for this comment. In the new version of the manuscript, recent studies were cited including the reviewer's suggestion.

Revised manuscript: (line 46) "Nowadays, the impressive potential of natural products [4,5], namely marine natural products (MNPs) [6,7] is increasingly recognized as they can provide chemical novelty and novel mechanisms of action."

In introduction novelty of the study should be explained.

Author´s response: We thank the reviewer for this observation. In the new version of the manuscript, the novelty statement was included in the introduction.

Revised manuscript: (line 106) "As far as we know, the chemical space occupied by the studied MPs was mapped for the first time using molecular descriptors and fingerprints and framed according to the drug-likeness concepts and to an antibacterial potential ranking."

Line 107 correct the sentence.

Author´s response: We thank the reviewer for this observation. This sentence was corrected in the new version of the manuscript.

Revised manuscript: (line 111) "Of these 246 MPs, 218 (89%) were isolated directly from marine sources, while 28 (11%) were obtained through semi-synthesis from a marine product and by recurring to simple reactions (acetylation [27–29], methylation [27,30], dehydration [30,31], or redox reactions [30,31])."

Materials and methods do not cover complete details. The authors are directed to provide proper details with links of the software and databases. Also provide scheme or graphical representation of the materials and methods data collection.

Author´s response: We understand the reviewer´s comment. In the new version of the manuscript, links of the software and databases were included as well as a scheme (Figure 2) to represent the data collection/analysis methodology.

Revised manuscript: (line 108) "Figure 2 depicts the schematic representation of the data collection and the tools used for the different calculations and analyses."

(line 459) "The literature survey was executed using Scopus (https://www.scopus.com/ accessed on 10 March 2023), Web of Science (https://www.webofscience.com accessed on 10 March 2023), and Google Scholar (https://scholar.google.com accessed on 10 March 2023) considering original research papers published up to June of 2022."

(line 467) "Molecular descriptors were calculated using the RDkit (release 2021_03_5 Q1 2021, https://www.rdkit.org/ accessed on 1 March 2023) and SwissADME [20]. Data analysis was performed in Python 3.8.16, using the RDkit 2021.03.5, Pandas 1.5.2, Numpy 1.23.5, Matplotlib 3.7.0, Seaborn 0.12.2, and Sci-kit learn 1.2.1 libraries. These tools were installed through Conda (https://docs.conda.io/projects/conda accessed on 3 January 2023). The code for the analysis is accessible at https://github.com/jxsoares. The chemical structures were drawn with MarvinSketch 22.22 (https://chemaxon.com accessed on 20 November 2022)."

The study lack discussion.

Author´s response: We understand the reviewer´s comment. Regarding the description of the chemical space, we think the level of discussion provided in the manuscript fulfils the goal of the study. Our intent was to provide a large overview of the molecular features present in these marine polyketides. Regarding the antibacterial activity analysis, we agree with the reviewer's comment. However, the large chemical diversity present in this dataset and the consequent possibility of encoding different mechanisms of action hampers a more detailed analysis. We gather this dataset as the first step of an antibacterial identification program. Currently, we are working on the synthesis of marine-inspired polyketides. We think that soon we will be able to provide a more detailed analysis of the antibacterial potential and action of these polyketides.

Conclusion looks like summary. Conclusion should be based on results providing study gap and future perspective.

Author´s response: We would like to thank Reviewer 2 for rising this important aspect. In the new version of the manuscript, we thoroughly revised the conclusions section.

Revised manuscript: (line 485) "PCA of the studied molecular descriptors revealed negative correlations between log P and log S, between log S and size, and between log P and TPSA. PCA also showed positive correlations between log S and TPSA and between MW and RB."

(line 488) "The identified MP have been sorted into clusters accordingly to their molecular similarity. The large number of clusters obtained using a loose threshold (threshold of 0.5 in the Butina clustering algorithm) highlights the large structural diversity present within this dataset. The analysis of natural products is usually performed with scaffold-based classification. However, our cluster-based classification, which is based on molecular fingerprints rather than on scaffold description, provided better discrimination between active and not active compounds than the conventional scaffold-based classification. The large chemical diversity present in the dataset was also evidenced by the TMAP which shows the common root between all MP and the numerous spanning branches encoding the different chemical structures. Coloring the TMAP map with our antibacterial potential ranking enabled the easy visualization of the regions of the chemical space containing the most promising compounds. Within the very promising compounds, four tricyclic polyketides were identified which exhibit good antibacterial activity and good drug-like properties. It should be mentioned that in this analysis we considered a broad notion of antibacterial activity based only on MIC values. The compounds identified as very promising might have different mechanisms of action and/or different potency for different bacteria strains. Independently of the intricacies of the antibacterial activity, these compounds constitute promising starting points for the design and synthesis of novel molecules which could expand our understanding of the antibacterial action of these marine natural products."

Reviewer 3 Report

The subject and the provided results are interesting and the authors have presented much information pertaining to their work; however, minor revision need to improve the quality of the manuscript. Therefore, I would like to make following comments for authors to revise their MS.

- Less attention has been paid to the mechanism (s) of marine Polyketides as an antibacterial agents in this MS. I suggest to comprehensively describe and  present a schematic Figure(s) related to general antimicrobial resistance mechanisms and their related key features. Describe how these compounds may act. 

As a new source of antibiotics, these compounds may also face the loss of antibacterial ability in the future. Which molecular features of marine polycetides do you think are the most crucial features to make it last longer in the medicine basket? Do you believe that you studied all of the molecular features in this MS? or there are a few other important ones missed? 

- English language needs some improvement.

Author Response

At first, we would like to express our gratitude to you for the time dedicated to our manuscript and the valuable comments which improved the quality of our paper. The manuscript was thoughtfully revised and modified according to the reviewer's recommendations.

Reviewer 3 comment´s:

The subject and the provided results are interesting and the authors have presented much information pertaining to their work; however, minor revision need to improve the quality of the manuscript. Therefore, I would like to make following comments for authors to revise their MS.

- Less attention has been paid to the mechanism (s) of marine Polyketides as an antibacterial agents in this MS. I suggest to comprehensively describe and present a schematic Figure(s) related to general antimicrobial resistance mechanisms and their related key features. Describe how these compounds may act.

Author´s response: We thank the reviewer for this observation. The four main antimicrobial resistance mechanisms are: (1) limiting uptake of a drug; (2) modifying an antibiotic compound; (3) inactivating a drug; (4) active drug efflux. Ref: doi: 10.3934/microbiol.2018.3.482. Several MPs with great antibacterial activity have been described in the literature. However, their mode of action has not been investigated in detail. Nevertheless, it is possible to rationalize how these natural compounds could act as a comparison with related compounds found in other natural sources or obtained by synthesis. Several xanthones derivatives that presented activity in efflux pumps as efflux pump inhibitors in Staphylococcus aureus (Ref: doi.org/10.3390/antibiotics10050600). Moreover, it was reported that the mechanism of action of xanthones is also related to the ability to partition the bacterial cell membrane and has a greater relationship with Gram-positive bacteria (ref: doi.org/10.3390/antibiotics12040645). Benzophenone derivates have shown good binding to the transpeptidases (ref: doi.org/10.3390/cimb45010007) and good DNA binding affinity, although mechanistic studies revealed that the antibacterial activity of these molecules is due to membrane depolarization of bacterial cells (Ref: 10.1021/jm900519b). Diphenyl ethers derivatives have shown strong interaction with E. coli DNA gyrase B (ref: 10.2147/AABC.S323657). Concerning the anthraquinone moiety, there are examples of anthraquinone derivates that have affinity to penicillin-binding protein 2a (ref: 10.1016/j.bioorg.2019.103515) and anthraquinone derivates which show a mechanism of action involving an increase in the levels of superoxide anion O2•− and/or singlet molecular oxygen 1O2 (ref: 10.1016/j.jphotobiol.2010.09.009). Bering in mind the structural similarity with some of marine polyketides present in this study, it is reasonable to assume that some of the compounds present here could have a similar mechanism of action. However, for a detailed analysis of their interaction and their mechanism of action, in silico or in vitro studies must be performed to obtain precise information. This vision was included in the new version of the manuscript.

Revised manuscript: (line 347) "Despite the promising antibacterial activity reported for MP, their mode of action has not been investigated in detail. Bearing in mind the structural similarity between these marine polyketides and terrestrial or synthetic polyketides, it is reasonable to assume that they could share similar modes of action. Natural and synthetic polyketides bearing the four studied scaffolds have been shown to act on efflux pumps [46], bacterial cell membrane [46,47], DNA gyrase B [48], transpeptidases [49], and through reactive oxygen species [50]. As the studied MP can show the different mechanisms of action, we focused our study on the de facto expressed antibacterial activity."

- As a new source of antibiotics, these compounds may also face the loss of antibacterial ability in the future. Which molecular features of marine polycetides do you think are the most crucial features to make it last longer in the medicine basket? Do you believe that you studied all of the molecular features in this MS? or there are a few other important ones missed?

Author´s response: The most important features are lipophilicity and water solubility. Water solubility is crucial because the drug must be dissolved at the site of absorption. Lipophilicity is crucial because it is the best predictor of membrane permeability. These two features affect deeply the potency and ADMET profile of a drug. In fact, the Biopharmaceutics Classification System differentiate drugs on the basis of their solubility and permeability. While poor permeability or poor solubility can be addressed along the drug development process, compounds showing simultaneously poor permeability and poor solubility have reduced chances of reaching the market. In the case of the studied marine products, in general, they showed good permeability and poor solubility. In this manuscript, we focused on the molecular features that are the most relevant ones (ref: 10.1038/srep42717). However, they are not the only ones. Efficiency is a very important feature as it allows us to infer the binding efficiency of a certain compound to its target. This feature is particularly important during the discovery phase as a selection criterion (in the presence of more than one compound with good potency, the selected one should be the most efficient). Ligand efficiency is usually expressed as lipophilic efficiency, defined as the pIC50 of interest minus the LogP of the compound, or as ligand efficiency, defined as binding energy (ΔG) to the number of non-hydrogen atoms of the compound. In this manuscript, these molecular descriptors were not applicable as the marine polyketides interact with different targets.

Reviewer 4 Report

The authors have chosen and analyzed a big set of natural polyketides, benzophenones, diphenyl ethers, anthraquinones, and xanthones isolated from different marine microorganisms for their potential antibacterial activity. For this dataset, molecular descriptors, and "fingerprints" were calculated. The polyketides have been set to 76 groups (clusters) accordingly to their molecular similarity, using the Butina clustering method, and into visualization trees map, using the TMAP unsupervised machine learning method. 

It has been shown that the discrimination between active and not active compounds was better with cluster-based classification than with scaffold-based classification. TMAP was also used to sort MPs according to their molecular similarity. The authors have identified four tricyclic polyketides exhibiting good antibacterial activity and good drug-like properties. 

The manuscript is well-written and looks like a review because summarizes a big amount of information on marine natural polyketides. I would advise the acceptance of this manuscript after some corrections mainly in the ESI file:

1. The Reference section which should contain 72 references (according to data in Table S1) is completely lost in ESI. It should be added.

2. The page numbers in Content given on page 2 of ESI are wrong and need to be corrected (e.g. Figure S2 is on page 96 but not 97 etc.).

3. The references in the main text should be provided in accordance with the Journal rules. 

4. Figure 2d page 4 of the manuscript) should be made more contrasted because it is hard to understand this figure in its present state. 

I wish authors good luck in future investigations!

From the reviewer's viewpoint, English is good enough, and, maybe, only minor editing of the English language is required.

Author Response

At first, we would like to express our gratitude to you for the time dedicated to our manuscript and the valuable comments which improved the quality of our paper. The manuscript was thoughtfully revised and modified according to the reviewer's recommendations.

Reviewer 4 comment´s:

The authors have chosen and analyzed a big set of natural polyketides, benzophenones, diphenyl ethers, anthraquinones, and xanthones isolated from different marine microorganisms for their potential antibacterial activity. For this dataset, molecular descriptors, and "fingerprints" were calculated. The polyketides have been set to 76 groups (clusters) accordingly to their molecular similarity, using the Butina clustering method, and into visualization trees map, using the TMAP unsupervised machine learning method.

It has been shown that the discrimination between active and not active compounds was better with cluster-based classification than with scaffold-based classification. TMAP was also used to sort MPs according to their molecular similarity. The authors have identified four tricyclic polyketides exhibiting good antibacterial activity and good drug-like properties.

The manuscript is well-written and looks like a review because summarizes a big amount of information on marine natural polyketides. I would advise the acceptance of this manuscript after some corrections mainly in the ESI file:

  1. The Reference section which should contain 72 references (according to data in Table S1) is completely lost in ESI. It should be added.

Author´s response: We would like to thank Reviewer 4 for rising this important aspect. This was a lapse, which was amended in the new version of the manuscript.

  1. The page numbers in Content given on page 2 of ESI are wrong and need to be corrected (e.g. Figure S2 is on page 96 but not 97 etc.).

Author´s response: We thank the reviewer for this observation. This was a lapse, which was amended in the new version of the manuscript.

  1. The references in the main text should be provided in accordance with the Journal rules.

Author´s response: We thoroughly checked and inserted the references in accordance with the journal rules.

  1. Figure 2d page 4 of the manuscript) should be made more contrasted because it is hard to understand this figure in its present state.

Author´s response: We thank the reviewer for this observation. The colors of the mentioned figure were adjusted to improve its contrast.

Round 2

Reviewer 1 Report

The authors revised the manuscript according the suggestions. I think it is acceptable for publication in Molecules.